# Identification of DCAF1 by Clinical Exome Sequencing and Methylation Analysis as a Candidate Gene for Autism and Intellectual Disability: A Case Report

**DOI:** 10.3390/jpm12060886

**Published:** 2022-05-27

**Authors:** Jeffery L. Clothier, Amy N. Grooms, Patricia A. Porter-Gill, Pritmohinder S. Gill, G. Bradley Schaefer

**Affiliations:** 1Psychiatric Research Institute, University of Arkansas for Medical Sciences, Little Rock, AR 72205, USA; agrooms@uams.edu; 2Arkansas Children’s Research Institute, Little Rock, AR 72202, USA; portergillpa@archildrens.org; 3Department of Pediatrics, University of Arkansas for Medical Sciences, Little Rock, AR 72202, USA; psgill@uams.edu; 4Genetics and Pediatrics, University of Arkansas for Medical Sciences, Little Rock, AR 72202, USA; schaefergb@uams.edu

**Keywords:** autism spectrum disorders (ASDs), exome sequencing, methylation, DCAF1

## Abstract

Autism spectrum disorder (ASD) comprises a heterogeneous group of neurodevelopmental disorders and occurs in all racial, ethnic, and socioeconomic groups. Cutting-edge technologies are contributing to understanding genetic underpinnings in ASD. The reported patient is a 32-year-old male and as an infant was noted to have microcephaly, hypospadias, pulmonary vascular anomaly, and small stature. He was diagnosed with Cornelia De Lange Syndrome (CDLS) at that time based on the clinical features. As a child, he had autistic features and intellectual disabilities and as diagnoses with autism and intellectual disability. He was referred as an adult to our neurodiversity clinic and a full exome trio sequencing with reflex to mitochondrial genes identified a de novo variant of uncertain significance in a candidate gene, *DCAF1*. The specific variant was c.137 C > T (p.Thr46Ile) in exon 4 in the *DCAF1* gene. In silico analysis supports a deleterious effect on protein structure/function. *DCAF1* participates with DDB1 and CUL4 as a part of the E3 ubiquitin ligase complex. The E3 ligase complex has been associated with a syndromic form of X-linked intellectual disability. The DDB1/CUL4 E3 ubiquitination complex plays a role in methylation-dependent ubiquitination. Next, a methylation study identified a signature similar to the methylation pattern found in X- linked intellectual disability type 93. This is associated with variants of the *BRWD3* gene, which is linked with the functioning of the DDB1/CUL4 E3 ubiquitination complex. Taken together, this suggests that the *de novo DCAF1* variant may be a newly identified molecular cause of autism and intellectual disability.

## 1. Introduction

Autism spectrum disorder (ASD) is a group of complex neurodevelopment disorders [1] which are characterized by persistent deficits in social communication and interaction as well as restricted, repetitive patterns of behavior. CDC estimates about 1 in 44 children have been identified with autism spectrum disorder (https://www.cdc.gov/ncbddd/autism/data.html, accessed on 27 March 2022) and 31% of children with ASD have an intellectual disability (https://www.autismspeaks.org/autism-statistics-asd, accessed on 27 March 2022). Recently, whole exome sequencing (WES) identified 102 high-confidence ASD-associated genes [2].

Trio-based exome sequencing provides insight into the contribution of de novo alterations in ASD incidence. It has increased the diagnostic yield in the evaluation of complex developmental disorders such as autism. Patients with psychiatric conditions in addition to autistic features were significantly more likely to receive positive results compared with patients without these features [3,4]. The candidate causal ASD-related genes with novel de novo mutations are identified, are often described by their function and relationship to the neurological disease. The relatively rare variants may not have been reported previously and this creates a situation of diagnostic uncertainty. Variants of uncertain significance (VUS) are often reported and attribution of a specific individual’s case to a VUS is sometimes possible.

DNA methylation is an epigenetic regulatory mechanism and affects gene expression at the transcriptional level [5]. Aref-Eshghi and colleagues [6] developed a machine learning model using genome-wide DNA methylation data from blood to predict 14 different Mendelian syndromes with neurodevelopmental presentations. Epigenetic chromatin dysregulation is a mechanism in the group of neurodevelopmental disorders known as chromatinopathies. These conditions are caused by the genetic alterations of various components of the epigenetic machinery [7]. Genetic syndromes often present with overlapping clinical features. Inconclusive genetic findings can confound accurate diagnosis and clinical management. A number of genetic syndromes have been shown to have unique genomic DNA methylation patterns. Methylation patterns may be useful for the interpretation of ambiguous genetic test results [6]. Cumulative DNA methylation patterns occurring at multiple CpG dinucleotides across the genome are intricately associated with many human traits and disease statuses.

We report here a case of a previously unreported VUS c.137 C > T (p.Thr46Ile) in a candidate gene *DCAF1*, which was associated with an abnormal methylation pattern similar to another known chromatinopathy.

## 2. Materials and Methods

This study was approved by the University of Arkansas for Medical Sciences Investigational Review Board (IRB# 274278) as NOT human subject research as defined in 45 CFR 46.102 and therefore it does not fall under the jurisdiction of the IRB review process. The following describes the molecular analyses on consented patient and parent samples for trio clinical exome sequencing and methylation.

### 2.1. Clinical Exome Sequencing

Exome sequencing (ES) approach selectively sequences the coding region (exons), and has become more valuable in the clinical setting and has contributed to earlier diagnosis, improved treatment, and prognosis [8]. Patient’s and parents’ blood samples were sent to GeneDx (https://www.genedx.com/, accessed on 20 December 2020) and genomic DNA was extracted and enriched for the complete coding regions and splice site junctions for most genes of the human genome using a proprietary capture system developed by GeneDx for next-generation sequencing with CNV calling (NGS-CNV).

The enriched targets were simultaneously sequenced with paired-end reads on an Illumina platform. Bidirectional sequence reads were assembled and aligned to reference sequences based on NCBI RefSeq transcripts and Human Genome build GRCh37/UCSC hg 19. Using a custom-developed analysis tool (XomeAnalyzer), the data were filtered and analyzed to identify sequence variants, and most deletions and duplications involving three or more coding exons [9]. Reported clinically significant variants were confirmed by an appropriate orthogonal method in the proband.

### 2.2. Methylation Analysis

The results of clinical exome sequencing were reviewed and the genetic counselor recommended a methylation profile of the patient. For the methylation study, bisulfite-treated DNA was isolated from peripheral blood and profiled using the Illumina MethylationEPIC BeadChip (https://www.illumina.com/products/by-type/microarray-kits/infinium-methylation-epic.html, accessed on 6 November 2021) array, which targets 862,927 CpG sites across the genome. The signal intensity of patient hybridized DNA was compared to reference DNA of healthy and affected cohorts. The signal intensity of the hybridized DNA from the patient sample was compared to reference DNA of healthy and affected cohorts. Methylation analysis was performed using model prediction algorithms designed by ABMGG-certified Clinical Molecular Geneticists of London Health Sciences Centre (Toronto, ON, Canada). Data processing, bioinformatics analysis, and epigenetic signature assessment were performed at the London Health Sciences Molecular Diagnostic Laboratories using the LHSC EpiSign™ Knowledge Database (London, ON, Canada) as per previous methods modifications [6,10,11,12]. The EpiSign Knowledge Database contains thousands of reference samples including a growing list of genetic conditions involving genetic developmental and intellectual delay and disabilities.

## 3. Results and Discussion

Based on a phenotypic exam, the male patient, as an infant, was diagnosed with Cornelia De Lange Syndrome (CDLS) and was noted to have microcephaly, hypospadias, pulmonary vascular anomaly, and small stature. During childhood he was diagnosed with autism and intellectual disability. He had relatively frequent behavioral outbursts and was incompletely responsive to behavioral management techniques such as stimulus control and contingency management. At the Children’s Hospital Outpatient System, the patient had been managed with aripiprazole, clomipramine, valproic acid with as required lorazepam. He had previous treatment with fluoxetine, sertraline, risperidone, and clonidine as an adult, with partial responses.

The patient now is 32 years old and entered the outpatient psychiatric services at the University of Arkansas for Medical Science (UAMS), Psychiatric Research Institute (PRI) for aggressive outbursts, oppositional defiant disorder, autism, intellectual disability, and obsessive-compulsive disorder. The patient was referred to the Neurodiversity Clinic at PRI for consultation with both a geneticist and senior psychiatrist for treatment recommendations. Clinical examination showed dysmorphic features such as short stature, red hair, microcephaly, short palpebral fissures, telecanthus, fleshy prominent helix and antihelix, flat philtrum, synophrys, and metopic forehead with a prominent nose.

Behaviorally, the patient had minimal verbal expression, impaired attention, but was able to understand spoken language and follow simple commands. The patient was unable to participate in cognitive examination beyond simple motor tasks and behaviorally bound to specific rituals including reactions to specific words. His aggressive behaviors were typically sudden and the family had adapted and used stimulus control to manage behaviors as they learned some of his specific triggers.

It was felt that some of patient’s features were not typical of those with Cornelia De Lange Syndrome (CDLS) and an exome trio sequencing with reflex to mitochondrial genes was performed. The results of the exome sequencing described a de novo variant of uncertain significance (VUS) in a candidate gene, *DCAF1* (NM_014703.2). The specific variant was c.137 C > T (p.Thr46Ile) in exon 4 in the *DCAF1* gene and sequencing data show no indication of multi-exon deletion or duplication. This DCAF1 variant had not been reported before, and in silico analysis supports a deleterious effect on protein structure/function. To more fully evaluate the potential effects of this variant, a methylation study was undertaken on this patient. The results were interpreted as consistent with X-linked intellectual disability type 93 (XLID-93). XLID-93 is associated with a protein coding gene *BRWD3*(Bromodomain, and WD-repeat domain-containing protein), by massive parallel sequencing of 407 families with X-linked mental retardation, Vulto-van Silfhout et al. (2015) identified CUL4B mutations in affected members of 8 families (2.0%). Subsequent screening of 29 patients with malformations of cortical development identified CUL4B mutations in 3 patients from 2 families. Ten different mutations were identified in the 10 families, including 5 truncating mutations, 2 splice site variants, an in-frame deletion, an in-frame duplication, and a missense variant by massive parallel sequencing of 407 families with X-linked mental retardation, Vulto-van Silfhout et al. (2015) identified CUL4B mutations in affected members of 8 families (2.0%). Subsequent screening of 29 patients with malformations of cortical development identified CUL4B mutations in 3 patients from 2 families. Ten different mutations were identified in the 10 families, including 5 truncating mutations, 2 splice site variants, an in-frame deletion, an in-frame duplication, and a missense variant which is a member of the DDB1/CUL4 E3 ubiquitin ligase complex [13]. Truncating mutations in the *BRWD3* gene were identified in 2 of 250 probands [13] and the affected males had macrocephaly with prominent forehead. Additionally, by massive parallel sequencing of 407 families with syndromic X-linked intellectual disability (XLID), mutations in *CUL4B* gene were identified in affected members of 8 families (2.0%) [14]. Subsequent screening of 29 patients with malformations of cortical development identified *CUL4B* mutations in 3 patients from 2 families. *CUL4B* variants which were identified included 5 truncating mutations, 2 splice site variants, an in-frame deletion, an in-frame duplication, and a missense variant [14].

*DCAF1* was initially discovered as a target for the human immunodeficiency virus accessory protein [15] and is a protein-coding gene on chromosome 3 with 32 exons. The normal targets and function of the CRL4 substrate receptor protein DDB1–Cul4-associated factor 1 (DCAF1; also known as VprBP) have remained elusive. Recent studies have begun to shed light on the physiologic role of DCAF1. DCAF1 participates with DDB1 and CUL4 as a part of the E3 ubiquitin ligase complex [6]. The DDB1/CUL4 E3 ubiquitination complex plays a role in methylation-dependent ubiquitination. Protein ubiquitination is a posttranslational modification that involves the covalent binding of a protein called ubiquitin to target proteins. Protein ubiquitination is a posttranslational modification that plays an integral role in diverse cellular functions. In the CNS, ubiquitination mediates many forms of synaptic plasticity, which ultimately affect learning, memory, and behavior. There are an estimated 600–700 E3 ligase genes. Several variants in E3 ligase genes have been observed in multiple neurological conditions; many of which are considered rare diseases [16]. Rare neurological disorders (RNDs) are a subtype of neurological diseases that represent 50% of all rare diseases, affecting fewer than 200,000 people in the United States.

There are seven members of the cullin family of E3 ligases. Each member interacts with specific adapters by acting as a scaffold. The adapters confer substrate specificity by acting as a recognition site for the targets. The CUL4 family members including CUL4A and CUL4B utilize more than 50 WD40-containing adapters referred to as DCAFs. Loss-of-function mutations in *CUL4B* gene have been reported in patients with X-linked intellectual disability [17,18,19,20] characterized by macrocephaly, aggressive outbursts, tremors, seizures, central obesity, and hypogonadism. The *DCAF1* gene has been shown to be integral as the recognition site for monomethylated sites that will undergo ubiquitination [21]. The role of histone modification is supported by studies of X-linked mental retardation (XLMR). At least seven proteins that have been identified as mutated in XLMR are potential methyl modifying enzymes or methyl binding proteins, including MECP2, JARID1C/SMCX95, PHF6, PHF8, BCOR, and BRWD3 [15]. DNA methylation signatures in autism have been reported [22] and recent studies show a DNA methylation signature in autism associated with Mecp2 [23] and CHD8 [24]. Wang and colleagues [25] report on DNA methylation signature with nonsyndromic nature of intellectual disability associated with *ZNF711*.

In our patient, it is interesting that the methylation pattern was similar to that of another chromatinopathy caused by a member of the DDB1-Cul E3 ligase complex [7]. The interpretation of methylation patterns in congenital disorders requires further study. The overlap of methylation patterns can be a basis for uncertain or incorrect classifications and that using DNA methylation for disease classifications should be performed with simultaneous consideration of all of the known methylation patterns [6,7]. DNA methylation patterns have the potential to shed light on the molecular mechanisms and assist in the diagnosis of congenital disorders. It may be helpful to consider the methylation patterns of other variants within the DDB1-Cul E3 ligase complex for similarities.

The rapid development of next-generation sequencing [26] and methylation analysis [22] are providing deeper understanding of molecular architecture of de novo variation in neurodevelopmental disorders; specifically, autism and intellectual disability [26,27]. These approaches may help define a unique endophenotype [26] at the genetic level and aid in clinical stratification of autistic patients with distinct de novo variations as the basis for guiding the diagnostic and therapeutic interventions.

## 4. Conclusions and Future Perspective

De novo variants are major contributors to the risk of autism [2,26], and this study identified a de novo variant c.137 C > T (p.Thr46Ile) in exon 4 of *DCAF1*. It is interesting that this patient’s methylation pattern was similar to that of another chromatinopathy caused by a member of the DDB1-Cul E3 ligase complex. The methylation pattern provided an additional helpful diagnostic aid to understand the clinical impact of a new variant of uncertain significance. It may be helpful to consider the methylation patterns of other variants within the DDB1-Cul E3 ligase complex for similarities. However, further research is needed to expand our current episignature databases in order to improve the sensitivity of examining methylation patterns [6,7]. Analyzing DNA methylation patterns has the potential to improve our understanding of molecular mechanisms and assist in the diagnosis of congenital disorders.

As the findings from this *DCAF1* research are novel, future studies with loss-of-function or gain-of-function approaches will give clues to clinical significance of this variant in exon 4 for the risk of autism and intellectual disability. It is highly likely that improved “Omic” approaches for next-generation sequencing and methylation will keep transforming the clinical diagnostics and therapeutics field, for genetically complex disorders such as autism and intellectual disability.

## Data Availability

Data is available upon request.

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
