# Peer review of "Identification of DCAF1 by Clinical Exome Sequencing and Methylation Analysis as a Candidate Gene for Autism and Intellectual Disability: A Case Report"

_jpm, 2022, doi:10.3390/jpm12060886_

Round 1

Reviewer 1 Report

The authors present a case report of an individual with a clinical diagnosis of autism and WES identified a de novo missense variant in DCAF1. This gene has not been so far associated with a Mendelian disorder. Further, they used methylation profiling test and identified a profile match with BRWD3, another member of the DDB1/CUL4 E3 ubiquitin ligase. This report is interesting as they followed up a de novo missense variant in a novel candidate gene using methylation profiling. However, the manuscript in the current version is not acceptable for publication and as it lacks several data including  variant details (below) and phenotypic comparison of features seen in individuals with BRWD3 causative variants and the current patient. Further, there is scope for making the manuscript text more readable as multiple sentences are unclear with lack of complete information.

Comments

  1. “DCAF1 participates with DDB1 and CUL4 as a part of the 20 E3 Ubiquitin ligase complex and is noted to cause a syndromic form of X-linked intellectual disability.”

In this sentence it is unclear which gene is being referred to cause a syndromic form of X-linked intellectual disability? Please revise the sentence for clarity.

  1. Variant details are missing : whether the variant was seen in population databases (GnomAD, TOPMED etc)? What is the transcript ID using which the variant was annotated – NM_?
  2. Is there any hint on the molecular mechanism of disease (loss of function vs gain of function) based on the methylation profile data ? The link between the BRWD3 and DCAF1 needs to be further discussed.
  3. “The reported patient is a 32 15 year old male with a history of a clinical diagnosis of DeLange Syndrome based on a phenotypic 16 exam as an infant.”

Are the authors referring here to Cornelia De Lange syndrome ? If so,  for the benefit of wider audience it might be better to change it to CDLS.

  1. Was the clinical laboratory which reported the WES results, requested to rule out any potential variants in BRWD3 in this patient sample, since the methylation profiles matched?
  2. Were large autism/ID databases reviewed for any other potential DN DCAF1 variants? or the gene added to GeneMatcher?
  3. A comment on the genetic counselling of the patient, if done,  will add to the case report.

Author Response

Thank you for your comments and suggestions. It has helped us to improve the clinical report. We have tried to make the text more readable, by repharasing many sentences throughout the manuscript and added pertinent references.

Comments

  1. “DCAF1 participates with DDB1 and CUL4 as a part of the 20 E3 Ubiquitin ligase complex and is noted to cause a syndromic form of X-linked intellectual disability.”

In this sentence it is unclear which gene is being referred to cause a syndromic form of X-linked intellectual disability? Please revise the sentence for clarity.

Thank you for your comment. We have revised sentence per your suggestion.

It reads now: DCAF1 participates with DDB1 and CUL4 as a part of the E3 Ubiquitin ligase complex. The E3 ligase complex has been associated with a syndromic form of X-linked intellectual disability.  Line 21-23.

  1. Variant details are missing : whether the variant was seen in population databases (GnomAD, TOPMED etc)? What is the transcript ID using which the variant was annotated – NM_?

Thank you for your comment. This variant in exon 4 of DCAF1 is not listed in GnomAd and TOPMED.

The information on transcript ID for DCAF1 is NM_014703.2. This has been added in the manuscript at line 147.

  1. Is there any hint on the molecular mechanism of disease (loss of function vs gain of function) based on the methylation profile data ? The link between the BRWD3 and DCAF1 needs to be further discussed.

The loss of function or gain of function studies were not done and none have been reported in the literature. We have explained in the manuscript that mutations in BRWD3 involves macrocephaly (line 158), but our patient with mutation in DCAF1 had microcephaly (line 83). Future studies with loss-of-function/gain-of-function using miRNA/siRNA strategies will help to show the clinical significance of this DCAF1 variant. Line 219-221.

  1. “The reported patient is a 32 year old male with a history of a clinical diagnosis of DeLange Syndrome based on a phenotypic exam as an infant.” We have rephrased this to make it clear that currently the age of patient is 32 years old. We have corrected this at lines 15 and 76 to show the correct age. As an infant, the patient was diagnosed with Cornelia De Lange syndrome (CDLS).

Are the authors referring here to Cornelia De Lange syndrome ? If so,  for the benefit of wider audience it might be better to change it to CDLS. Thank you for your comment, we have changed it to CDLS (line 16 and 71) .

  1. Was the clinical laboratory which reported the WES results, requested to rule out any potential variants in BRWD3 in this patient sample, since the methylation profiles matched? BRWD3 (Bromodomain and WD Repeat Domain Containing 3) is a Protein Coding gene and clinical exome sequencing did not show mutations in BRWD3. The mutation in BRWD3 is associated with macrocephaly (line 157), and our patient had no macrocephaly features.  The report clearly shows the mutation in DCAF1 and that the patient hasmicrocephaly (line 71 and 158).
  2. Were large autism/ID databases reviewed for any other potential DN DCAF1 variants? or the gene added to GeneMatcher?

The GeneMatcher database does not have this gene in their database. Also we explored the Safari data base (https://gene.sfari.org/database/human-gene/) and DCAF1 is not listed.

  1. A comment on the genetic counselling of the patient, if done,  will add to the case report.

Thank you for this comment. Yes, a Genetic counselor recommended the methylation study, as well as a follow up on whole exome sequencing, to confirm whether new information on the variant analysis was added in the data base. See line 127-28.

Reviewer 2 Report

Thank you for the opportunity to review an interesting case report study on a de novo variant of uncertain significance in ASD, with associated methylation pattern.

I find the study and results relevant to the field, especially considering the increasing interest in the role of de novo mutations in ASD, and how these can be used for stratification of specific characteristics, like cognitive abilities.

I think, however, that the discussion needs to be expanded and a major effort should be made to discuss findings in the context of previous work on methylation and the role of de novo mutations in the genetic landscape of ASD.

Particularly, the authors refer to the DCAF1 as a newly identified molecular cause of intellectual disability, and this should be placed in the context of previous literature linking de novo mutations with intellectual abilities in the context of autism, as well as discussed in relation to the larger genetic landscape for ASD.

Similarly, the authors should expand on the contribution of these novel approaches to the diagnostic process, particularly in relation to ASD heterogeneity.

Author Response

Thank you for your suggestions and comments which were very helpful to improve the manuscript. Below are are our responses to you comments/suggestions.

Thank you for the opportunity to review an interesting case report study on a de novo variant of uncertain significance in ASD, with associated methylation pattern.

  1. I find the study and results relevant to the field, especially considering the increasing interest in the role of de novo mutations in ASD, and how these can be used for stratification of specific characteristics, like cognitive abilities.

 The de novo mutations can be definitely used for autism patient stratification. Once the de novo variant and methylation pattern are established, it can be used to give diagnostic and therapeutic intervention, and hence help stratify patients in different variant subgroups/endophenotypes. See lines 200-204.

  1. I think, however, that the discussion needs to be expanded and a major effort should be made to discuss findings in the context of previous work on methylation and the role of de novo mutations in the genetic landscape of ASD.

We have added this information on methylation and de novo variation in ASD. Please see lines 186-190.

  1. Particularly, the authors refer to the DCAF1 as a newly identified molecular cause of intellectual disability, and this should be placed in the context of previous literature linking de novo mutations with intellectual abilities in the context of autism, as well as discussed in relation to the larger genetic landscape for ASD.

Thank you for this comment. The genetic landscape of ASD is truly complex (reference 2 and 27). We have added the information on intellectual disability with ASD. Please see lines 153-158 and 179-181.

4. Similarly, the authors should expand on the contribution of these novel approaches to the diagnostic process, particularly in relation to ASD heterogeneity.

Thank you for this comment, Because we are limited by the 3000 word count, we have justified as much as we could add in results and discussion. Please see the final conclusion information on lines 201-207.

Round 2

Reviewer 1 Report

The manuscript has improved compared to the previous version, however it still needs several changes before it is acceptable for publication in JPM. 

Comments:

  • I suggest to modify the title to ““Identification of DCAF1 by Clinical Exome Sequencing and Methylation Analysis as a candidate gene for Autism: A Case Report”
  • Line 19-20“The specific variant was p.Thr46Ile (ACT>ATT): 19 c.137 C>T in exon 4 in the DCAF1 gene.”

The “(ACT>ATT)” part is redundant and instead the variant should be represented as c.137C>T (p.Thr46Ile) throughout the manuscript

  • Line 15-17 “The reported 15 patient is a 32 year old male and as an infant was given with a history of a clinical diagnosis of 16 Cornelia De Lange Syndrome (CDLS)”

This  sentence needs to be modified to also include the current clinical diagnosis and phenotype

  • If this individual has a clinical diagnosis of autism as well as ID, then the title and the last sentence of the abstract should include these two phenotypes.
  • It is unclear what the authors are trying to convey with these sentences (below). Are they referring to novel genes associated with ASD which are currently not reported in clinical exome sequencing?

Line 43-47 “Unique ASD-related genes recently 43 identified are often described as “nonclassical,” ASD genes with novel pathogenic se- 44 quence variants. The relatively rare variants may not have been reported previously and This creates a situation of diagnostic uncertainty.”

  • For the sentence below, are the authors trying to convey that, for some VUS variants with follow-up testing( if available), can resolve whether the variant is causal or negate it from being causal to the patient’s phenotype?

Line 46-47 “Variants of uncertain significance 46 (VUS) are often reported and a. Attribution of a specific individual’s case to a VUS is 47 sometimes possible.”

  • Please modify this sentence

Line 61-63 “We report here a case of a previ- 61 ously unreported VUS of a candidate gene, DCAF1, which was associated with an abnor- 62 mal methylation pattern similar to another known chromatinopathy.”

To

We report here a case of a previ- 61 ously unreported VUS in a candidate gene DCAF1, which was associated with an abnor- 62 mal methylation pattern similar to another known chromatinopathy.”

  • The case presentation part (entire 2.1, lines 71-90) needs to be moved to the Results and Discussion section. Infact, the authors can start the results section with the case presentation write-up.
  • Line 152 - “this. This was a de novo 152 variant.” This is redundant as it was mentioned above.
  • Line 199: “methylation patterns have a potential to understand the molecular mechanisms and assist 200 in the diagnosis of congenital disorders.”

Modify this to

methylation patterns have a potential to shed light on the molecular mechanisms and assist  in the diagnosis of congenital disorders.

  • Line 220: However, further research is need to expand our current episignature 220 databases in order to improve the sensitivity of examining methylation patterns [6,7, 7].

Needed?

Was the patient/ family consented for the WES & methylation testing ? 

Author Response

Thank you for your help to improve our submission. Please find below the responses to your comments and suggestions which were very helpful. We have changed the title to include autism and intellectual disability. We have accepted all your changes.

The manuscript has improved compared to the previous version, however it still needs several changes before it is acceptable for publication in JPM. 

Comments:

  • I suggest to modify the title to ““Identification of DCAF1 by Clinical Exome Sequencing and Methylation Analysis as a candidate gene for Autism: A Case Report”. Thank you for this suggestion, we have modified the title and it reads now as : 
  • Identification of DCAF1 by Clinical Exome Sequencing and Methylation Analysis as a Candidate gene for Autism and Intellectual disability: A Case Report”.
  •  
  • Line 19-20“The specific variant was p.Thr46Ile (ACT>ATT): 19 c.137 C>T in exon 4 in the DCAF1 gene.”  
  • Thank you for this suggestion, now we incorporated c.137C>T (p.Thr46Ile) in manuscript-see line 21, 63, 157. 221
  • The “(ACT>ATT)” part is redundant and instead the variant should be represented as c.137C>T (p.Thr46Ile) throughout the manuscript.
  • Line 15-17 “The reported 15 patient is a 32 year old male and as an infant was given with a history of a clinical diagnosis of 16 Cornelia De Lange Syndrome (CDLS)”  
  • We have include the clinical diagnosis per your suggestion. See lines 17-22. It reads as “The reported patient is a 32-year-old male and as an infant was noted to have microcephaly, hypospadias, pulmonary vascular anomaly, and small stature. He was diagnosed with Cornelia De Lange Syndrome (CDLS) at that time based on the clinical features. As a child, he had autistic features and intellectual disabilities and as diagnoses with Autism and Intellectual Disability. He was referred as an adult to our neurodiversity clinic and a full exome trio sequencing “
  • This  sentence needs to be modified to also include the current clinical diagnosis and phenotype.
    • If this individual has a clinical diagnosis of autism as well as ID, then the title and the last sentence of the abstract should include these two phenotypes.
    •  
  • Thank you we have incorporated this cahnge. See lines 4, 21, 30 and 223
  •  
  • It is unclear what the authors are trying to convey with these sentences (below). Are they referring to novel genes associated with ASD which are currently not reported in clinical exome sequencing?   
  • We apologize for the confusion. We have rephrased it to The candidate causal ASD-related genes with novel de novo mutations are identified, are often described by their function and relationship to the neurological disease. See lines 51-54.
  • Line 43-47 “Unique ASD-related genes recently 43 identified are often described as “nonclassical,” ASD genes with novel pathogenic se- 44 quence variants. The relatively rare variants may not have been reported previously and This creates a situation of diagnostic uncertainty.”
  • For the sentence below, are the authors trying to convey that, for some VUS variants with follow-up testing( if available), can resolve whether the variant is causal or negate it from being causal to the patient’s phenotype?YES, that is what we meant to convey.
  •  
  • Line 46-47 “Variants of uncertain significance 46 (VUS) are often reported and a. Attribution of a specific individual’s case to a VUS is 47 sometimes possible.”
  • Please modify this sentenceTo  
  • Thank you for this suggestion. We have changed it per your suggestions-please see lines 71-75.
  • We report here a case of a previ- 61 ously unreported VUS in a candidate gene DCAF1, which was associated with an abnor- 62 mal methylation pattern similar to another known chromatinopathy.”
  • Line 61-63 “We report here a case of a previ- 61 ously unreported VUS of a candidate gene, DCAF1, which was associated with an abnor- 62 mal methylation pattern similar to another known chromatinopathy.”
  • The case presentation part (entire 2.1, lines 71-90) needs to be moved to the Results and Discussion section. Infact, the authors can start the results section with the case presentation write-up. Thank you for your suggestion, we have moved this to Results and discussion. Lines ---138-160.
  •  
  •  
  • Line 152 - “this. This was a de novo 152 variant.” This is redundant as it was mentioned above. Ok we have deleted this sentence line 170.
  • Line 199: “methylation patterns have a potential to understand the molecular mechanisms and assist 200 in the diagnosis of congenital disorders.”methylation patterns have a potential to shed light on the molecular mechanisms and assist  in the diagnosis of congenital disorders. Thank you for this suggestion, we have incorporated this change -see lines 217-218.
  •  
  •  
  • Modify this to
  • Line 220: However, further research is need to expand our current episignature 220 databases in order to improve the sensitivity of examining methylation patterns [6,7, 7].    
  • For any genetic test at UAMS/ACH patient/family consent is required. Yes, the patient and family were consented for the tests. Line 80.
  • Was the patient/ family consented for the WES & methylation testing ? 
  • Needed changed to required line line 145; Thank you we corrected it to [6,7]
  • Needed?

Bottom of Form